# Zearalenone-14-Glucoside Is Hydrolyzed to Zearalenone by β-Glucosidase in Extracellular Matrix to Exert Intracellular Toxicity in KGN Cells

**DOI:** 10.3390/toxins14070458

**Published:** 2022-07-04

**Authors:** Haonan Ruan, Yunyun Wang, Yong Hou, Jing Zhang, Jiashuo Wu, Fangqing Zhang, Ming Sui, Jiaoyang Luo, Meihua Yang

**Affiliations:** Key Laboratory of Bioactive Substances and Resources Utilization of Chinese Herbal Medicine, Ministry of Education, Institute of Medicinal Plant Development, Chinese Academy of Medical Sciences & Peking Union Medical College, Beijing 100193, China; ruanhn@126.com (H.R.); cloud22wang@163.com (Y.W.); houyongyzs@sina.com (Y.H.); m13602175765@163.com (J.Z.); wjs_implad@163.com (J.W.); iszhangfq@163.com (F.Z.); suiming970315@163.com (M.S.)

**Keywords:** zearalenone-14-glucoside, UHPLC-ESI-MS/MS, β-glucosidase, metabolism, toxic release mechanism

## Abstract

As one of the most important conjugated mycotoxins, zearalenone-14-glucoside (Z14G) has received widespread attention from researchers. Although the metabolism of Z14G in animals has been extensively studied, the intracellular toxicity and metabolic process of Z14G are not fully elucidated. In this study, the cytotoxicity of Z14G to human ovarian granulosa cells (KGN) and the metabolism of Z14G in KGN cells were determined. Furthermore, the experiments of co-administration of β-glucosidase and pre-administered β-glucosidase inhibitor (Conduritol B epoxide, CBE) were used to clarify the mechanism of Z14G toxicity release. Finally, the human colon adenocarcinoma cell (Caco-2) metabolism model was used to verify the toxicity release mechanism of Z14G. The results showed that the IC_50_ of Z14G for KGN cells was 420 μM, and the relative hydrolysis rate of Z14G on ZEN was 35% (25% extracellular and 10% intracellular in KGN cells). The results indicated that Z14G cannot enter cells, and Z14G is only hydrolyzed extracellularly to its prototype zearalenone (ZEN) by β-glucosidase which can exert toxic effects in cells. In conclusion, this study demonstrated the cytotoxicity of Z14G and clarified the toxicity release mechanism of Z14G. Different from previous findings, our results showed that Z14G cannot enter cells but exerts cytotoxicity through deglycosylation. This study promotes the formulation of a risk assessment and legislation limit for ZEN and its metabolites.

## 1. Introduction

Due to the extensive contamination in food and feed and their multi-organ toxicity, mycotoxins have attracted the attention of researchers all over the world [1]. Previous studies have found that some mycotoxins can be modified with substances, such as sugar, sulfate, and amino acids. Currently, conjugates formed by residual mycotoxins in plant tissues, known as masked mycotoxins, are neither routinely screened for in food and feed nor regulated by legislation [2]. Some masked mycotoxins may be converted into prototype toxins in the gastrointestinal (GI) tract of humans or animals, resulting in toxicity similar to their original forms [3,4]. Therefore, the harm of masked mycotoxins cannot be ignored.

Masked mycotoxins have been studied by many researchers, Z14G widely exists in food and feed as a conjugated mycotoxin and has received a lot of attention [5]. Polish researchers detected 407 ± 61 µg/kg of Z14G in wheat flour [6]. Dall’Erta et al. verified that Z14G can easily undergo deglycosylation under the action of human colonic microbial flora to generate its prototype ZEN, thereby causing harm to human health [7]. In addition to the intestinal microbes, subsequent metabolic experiments also showed that Z14G was metabolized to varying degrees in animal serum, liver microsomes, and cells [8,9,10]. Research on the analysis and decontamination of Z14G in food and feed is also increasing. Nathanail et al. established a novel liquid chromatography–tandem mass spectrometric method for the simultaneous determination of major type A and B trichothecenes, zearalenone, and certain modified metabolites in Finnish cereal grains [11]. Experimental results from Faisal et al. indicated that cyclodextrin technology seemed a promising tool to improve the fluorescence analytical detection of Z14G as well as to decrease Z14G exposure through the removal of Z14G from aqueous solutions [12]. According to EU regulations, ZEN in food should not exceed 50 μg/kg, and ZEN in food for infants and young children should not exceed 20 μg/kg [13]. In order to establish a more comprehensive ZEN limit standard, the European Food Safety Authority (EFSA) put forward a series of recommendations for the promotion of research on metabolism and toxicology for the masked mycotoxins of ZEN [14]. Among them, there have been many reports on the metabolism and toxicity of α-ZEL and β-ZEL [15,16]. As one of the most important conjugated mycotoxins of ZEN, although Z14G had many metabolic studies [9,17,18], its toxicology studies were rarely reported.

As an estrogen receptor activator, the main effect of ZEN intake on human and animals is reproductive toxicity [19]. Recent studies have found that ZEN can induce apoptosis of mouse ovarian granulosa cells by activating caspase-3, Bcl2-associated x, and poly ADP-ribose polymerase [20]. Liu et al. reported that ZEN disrupts genome stability and inhibits growth of porcine granulosa cells via the estrogen receptors which may promote granulosa cells apoptosis [21]. This study aimed to study the toxicity of Z14G by using KGN cells, a human-derived cell line sensitive to ZEN. As a classic in vitro metabolism model, Caco-2 cells have been widely used to study the metabolism and distribution of mycotoxins [22,23]. Kohn B et al. studied the distribution and metabolism of modified mycotoxins of alternariol, alternariol monomethyl ether, and zearalenone in the Caco-2 cells model [24].

In this study, KGN cells were used as an in vitro toxicity model to determine the toxicity of Z14G, and the key factors of Z14G toxicity were clarified by administrating β-glucosidase and β-glucosidase inhibitors. The toxicity release mechanism of Z14G was further explored by using ultra-high-performance liquid chromatography–electrospray ionization-tandem mass spectrometry (UHPLC-ESI-MS/MS). Finally, the in vitro toxicity release mechanism was verified in Caco-2 cells. The aim of this study was to explore whether Z14G can directly enter cells to exert its own cytotoxicity, or whether Z14G can only exert ZEN cytotoxicity through hydrolysis into ZEN. In addition, we also studied the reaction site (intracellular or extracellular) of Z14G hydrolysis to ZEN.

## 2. Results

### 2.1. The Cytotoxicity of Z14G on KGN Cells

In order to determine the cytotoxicity of Z14G, we used KGN cells as an in vitro cytotoxicity model to explore the toxicity of Z14G on the reproductive system. MTT results showed that compared with the control group, after 24 h administration of high-dose (290–540 μM) Z14G, the KGN cells viability significantly reduced, and the IC_50_ was 420 μM. Compared with the ZEN administration group, the IC_50_ of Z14G on KGN cells was 5.25 times that of ZEN (Figure 1). Administration of 420 μM Z14G for 24 h caused significant changes in KGN cells morphology, thus promoting cell shrinkage and fragmentation (Appendix A).

### 2.2. Metabolism and Absorption of Z14G in KGN Cells

A previous study had shown that Z14G can be hydrolyzed into ZEN to exert toxicity in vivo [7]. In order to clarify the detailed metabolism of Z14G inside and outside KGN cells, we used UHPLC-ESI-MS/MS to determine the intracellular and extracellular contents of Z14G, ZEN, and its metabolites in KGN cells after 24 h administration of Z14G or ZEN (20 μM). The method validation results showed that the precision was <3%, and recovery was between 80–110% (Appendix A). After administration of 20 μM ZEN, it was metabolized into a small amount of α-ZEL and β-ZEL inside and outside cells, and the relative metabolism rate of α-ZEL (10%) was higher than that of β-ZEL (4%). In addition, ZEN, α-ZEL, and β-ZEL were able to enter cells to exert their cytotoxicity (Figure 2a,b). In extracellular experiment of KGN cells of 20 μM Z14G group, we observed that Z14G was extracellularly hydrolyzed to ZEN (25%) and further metabolized (Figure 2c); surprisingly, after administration of 20 μM Z14G, ZEN (10%) was detected in KNG cells, while Z14G was not detected intracellularly (Figure 2d). This result suggests that Z14G cannot enter cells. The results are helpful for investigating the toxicity release mechanism of Z14G in KGN cells.

### 2.3. Effects of β-Glucosidase and Its Inhibitors on the Toxicity and Metabolism of Z14G in KNG Cells

To further explore the toxicity release mechanism of Z14G in KGN cells, we performed β-glucosidase co-administration and CBE pretreatment experiments while administering Z14G to KGN cells. CBE has been widely used in the biomedical field as a β-glucosidase inhibitor [25]. MTT results showed that, compared with the control group, β-glucosidase administration and CBE pretreatment had no significant impact on the cell viability of KGN; compared with the Z14G group (50%), the cell viability of the β-glucosidase + Z14G group (10%) was significantly reduced (Figure 3). According to synchronous metabolic experiments, Z14G in the β-glucosidase + Z14G group was completely hydrolyzed into ZEN under the action of β-glucosidase (Figure 4a,b). In contrast, compared with the Z14G group, the cell viability of the CBE + Z14G group was significantly increased (Figure 4), and the LC-MS/MS analysis results showed that the pretreatment of CBE completely avoided the hydrolysis of Z14G. Z14G was not detected intracellularly, and ZEN was not detected either intracellularly or extracellularly (Figure 4c,d). The results confirmed the hypothesis that Z14G cannot enter cells (Figure 5), and Z14G has almost no cytotoxicity outside cells (Figure 3).

### 2.4. Metabolism and Absorption of Z14G in Caco-2 Cells: Verification of Z14G Toxicity Release Mechanism

The above experimental results indicated that Z14G was hydrolyzed into ZEN by β-glucosidase extracellularly, and ZEN enters KGN cells to exert cytotoxicity. To further verify the extensiveness and reliability of the toxicity release mechanism of Z14G, we used UHPLC-ESI-MS/MS to determine the metabolism and distribution of Z14G in Caco-2 cells (in vitro metabolism model). Consistent with the previous results, Z14G and its hydrolysate ZEN were simultaneously detected extracellularly (Figure 6a), while ZEN and its metabolites were detected, and Z14G was not detected inside Caco-2 cells (Figure 6d). Finally, we added Z14G to blank DMEM or the DMEM after culturing with Caco-2 cells for 24 h, to investigate the metabolism. The results showed that compared with Z14G in the blank DMEM co-incubation group, Z14G was hydrolyzed into ZEN in the DMEM after culturing with Caco-2 cells for 24 h (Figure 6b,c). These results demonstrated that Z14G is extracellularly hydrolyzed into ZEN.

## 3. Discussion

At present, conjugated mycotoxins with high detection rates and high attention mainly include Z14G and deoxynivalenol-3-β-glucoside (D3G). Although it has been proven that conjugated mycotoxins can exert their toxicity through hydrolysis into prototypes, their hydrolysis rate, toxicity dose, and toxicity release mechanism in human cells remain unknown. A previous study showed that Z14G (1 μM) was hydrolyzed to ZEN in Michigan Cancer Foundation-7 (MCF-7) cells, but there was almost no cytotoxicity [26]. The results of Alix Pierron et al. showed that D3G (10 μM) did not activate c-Jun N-terminal kinase (JNK) and mitogen-activated protein kinase (MAPKs) in treated Caco-2 cells and did not alter the viability and barrier function of cells, and no morphological damage was observed in the intestinal explants treated with D3G (10 μM) for 4 h [27]. Our study first confirmed the in vitro toxicity of conjugated mycotoxins, i.e., administration of 420 μM Z14G on KGN cells and incubation for 24 h can cause a significant decrease in cell viability. The high concentration of Z14G was used for cytotoxicity experiments because we needed to obtain the IC_50_ value of Z14G for KGN to determine the modeling dose of Z14G to induce KGN cytotoxicity. It is undeniable that our experimental results showed that in the absence of the participation of intestinal flora or β-glucosidase, Z14G has low toxicity or even no toxicity to KGN cells, but it does not mean that Z14G has no toxicity to cells after entering the human body. Our other experiments also demonstrated that Z14G was as toxic as ZEN in the presence of β-glucosidase. In addition, its toxicity is proportional to the hydrolysis rate of Z14G.

Our experimental results on the metabolism and distribution of Z14G in Caco-2 are different from previous reports [28]. Research by De N.M. et al. showed that D3G was not hydrolyzed to deoxynivalenol (DON) in the digestion model representing the upper part of the GI -tract, and D3G was not hydrolyzed to DON by the intestinal epithelial Caco-2 cells [23]. Similar to our study, Caco-2 cells did not absorb D3G, but Z14G was more easily hydrolyzed than D3G. Our study elucidates the toxicity release mechanism of Z14G, and indicates that Z14G was hydrolyzed extracellularly to ZEN by β-glucosidase to exert toxicity in cells.

Our study also proves that the presence of β-glucosidase was the most critical factor affecting the hydrolysis of Z14G. The hydrolysis of Z14G in the DMEM after culturing with Caco-2 cells for 24 h indicated that there was sufficient β-glucosidase in the extracellular matrix to partially hydrolyze Z14G during cell culture. At the same time, the pretreatment of CBE inhibited the metabolism and toxicity of Z14G, which further showed that Z14G itself was almost non-toxic. This was consistent with the previous report, which found that Z14G could not effectively bind and activate estrogen receptors through an in vitro/in silico integrated approach [10]. In other words, the hydrolysis efficiency of Z14G in different cells or gut microbes determines its toxicity, which is the reason why Z14G has different varying toxicity in different cells.

In vitro cell experiments can accurately and simply elucidate the toxicity release mechanism of Z14G in cells, but it cannot be simply equated with the metabolism and toxicity of Z14G when entering the human body. We believe that Z14G’s dissolution state, bioavailability, effect on the gut microbes, and other factors that differ from those of ZEN in the animal’s gastrointestinal tract should be fully considered. Therefore, we will further explore the toxicity mechanism of Z14G in vivo through animal experiments, especially the difference in toxicity between Z14G and ZEN, which may give us a more comprehensive and in-depth understanding of the toxicity mechanism of conjugated mycotoxins.

Although EFSA identified the assumptions that hydrolysis of ZEN conjugates in the intestinal tract was complete, that the estrogenic activities of ZEN and its phase I metabolites were additive, the limit standard for Z14G has not yet been perfected due to the lack of toxicological evidence. A recent risk assessment for modified ZEN pointed out that clarifying toxicology and re-evaluating the group tolerable daily intake (TDI) are the most pressing challenges associated with health risk assessment of modified mycotoxins [29]. Our study has clarified the cytotoxicity and toxicity release mechanism of Z14G and provides a theoretical basis for the modified ZEN legislation limit and risk assessment.

## 4. Conclusions

This study reported the in vitro cytotoxicity of conjugated mycotoxins for the first time and clarified the toxicity release mechanism of Z14G by measuring the cell viability of Z14G and the metabolism of Z14G in KGN cells. Through administration of the key enzyme β-glucosidase and preadministration of β-glucosidase inhibitor, we verified that Z14G was only hydrolyzed extracellularly by β-glucosidase to its prototype ZEN, which then entered cells to exert its toxic effects, while Z14G cannot enter cells. This mechanism was verified in the Caco-2 cells metabolism model. In conclusion, this study demonstrated the objective law that Z14G itself could not enter the cell, and proved that the reaction site of Z14G hydrolysis was outside cells. This study lays the foundation for the subsequent in vivo metabolism and toxicity study of Z14G, and it also provides mentalities for the study of other conjugated mycotoxins. More importantly, this study provides a theoretical basis for the legislation limit and risk assessment of modified ZEN.

## 5. Materials and Methods

### 5.1. Chemicals and Reagents

Z14G was purchased from Qiyun Biotechnology Co., Ltd. (Guangzhou, China), ZEN was purchased from Yuanye Biology Co., Ltd. (Shanghai, China), α-Zearalenol (α-ZEL) and β-Zearalenol (β-ZEL) were purchased from Pribolab (Qindao, China).

Dulbecco’s modified eagle medium (DMEM), penicillin/streptomycin (100×), fetal bovine serum (FBS), phosphate buffered saline (PBS), and trypsin were purchased from Corning (USA), and RIPA lysis buffer and thiazolyl blue tetrazolium bromide were purchased from Solarbio (Beijing, China).

β-Glucosidase (6 U/mg, Duoxi) was purchased from Jinsui Biotechnology Co., Ltd. (Beijing, China); Conduritol B epoxide was purchased from Med Chem Express (Shanghai, China).

### 5.2. Cell Culture and Treatment

Human ovarian granulosa cell (KGN) or human colon adenocarcinoma TC-7 cell (Caco-2) were purchased from the BeNa Culture Collection and routinely cultured in DMEM, supplemented with 10% FBS and 1% penicillin/streptomycin at 37 °C, pH 7.4, in an atmosphere of 5% CO_2_.

Groups were divided according to the following different administration methods:(1)Cytotoxic experiment

(i) Control group: DMEM; (ii) Solvent group: DMEM containing 1% methanol; (iii) ZEN group: ZEN (10, 20, 40, 80 and 160 μM) dissolved in DMEM containing 1% methanol; (iv) Z14G group: Z14G (290, 380, 420, 460 and 540 μM) dissolved in DMEM containing 1% methanol; (v) β-glucosidase + Z14G group: β-glucosidase (100 U·mL^−1^) + Z14G (420 μM) dissolved in DMEM containing 1% methanol; (vi) CBE + Z14G group: CBE (100 μM) pretreatment for 48 h + Z14G (420 μM) dissolved in DMEM containing 1% methanol. For ZEN or Z14G group, cells were incubated with ZEN or Z14G for 24 h. In β-glucosidase + Z14G group, cells were incubated with both β-glucosidase and Z14G for 24 h. In CBE + Z14G group, cells were incubated with CBE for 48 h, and then the old medium was discarded, and fresh medium containing Z14G was added, and the cells were further incubated for 24 h. Each experimental group was biologically repeated six times.


(2)Cell metabolism experiment


(i) Control group: DMEM; (ii) Solvent group: DMEM containing 0.05% methanol; (iii) ZEN group: ZEN (20 μM) dissolved in DMEM containing 0.05% methanol; (iv) Z14G group: Z14G (20 μM) dissolved in DMEM containing 0.05% methanol; (v) β-glucosidase + Z14G group: β-glucosidase (100 U·mL^−1^) + Z14G (20 μM) dissolved in DMEM containing 0.05% methanol; (vi) CBE + Z14G group: CBE (100 μM) pretreatment for 48 h + Z14G (20 μM) dissolved in DMEM containing 0.05% methanol; (vii) Z14G in the DMEM after culturing with Caco-2 cells for 24 h group: Z14G (20 μM) + the DMEM after culturing with Caco-2 cells for 24 h; (viii) Z14G in blank DMEM group: Z14G (20 μM) dissolved in DMEM containing 0.05% methanol. The incubation time of compounds and cells is the same as mentioned above. Each experiment group was biologically repeated six times.

### 5.3. Cell Viability Assay

Cell viability assay was performed as described previously [30]. KGN cells were plated in 96-well plates in DMEM in sextuplicate (5 × 10^4^ cells/well). Each cytotoxic group was administered according to Section 5.2—(1). MTT reaction mix (2.5 mM MTT, phenol-red free, FBS-free, FKS free, and Pen/Strep free DMEM) was added to each well and incubated for 4  h at 37  °C for MTT incorporation. To stop the MTT reaction, stop solution (2% dimethyl sulfoxide, 0.1 M glycine pH 11) was added to each well and incubated for 5 min, the plates were then read in a microplate reader at 570 nm wavelength, and the cell viability of each group was calculated.

### 5.4. Sample Preparation

KGN or Caco-2 cells were plated in 6-well plates in DMEM in sextuplicate (1.2 × 10^6^ cells/well). After treatment, the cell supernatant was collected, and the cells were washed three times with PBS, RIPA lysis buffer was added, and the cell-lysate was collected after shaking for 3 min. The cell supernatant and lysate were then extracted with 10 times (*v*/*v*) ethyl acetate. After centrifugation at 6500× *g* for 5 min at room temperature, the extract was collected and dried under a stream of nitrogen at 40 °C. Samples were then reconstituted with a mixture of methanol and water (50:50, *v*/*v*) containing 0.1% acetic acid, and passed through a 0.22 μm nylon filter before UHPLC-ESI-MS/MS analysis.

### 5.5. UHPLC-ESI-MS/MS Analysis and Quantification of Z14G and Its Metabolites

UHPLC-ESI-MS/MS analysis was performed as described previously [31]. The UHPLC system was operated using Shimadzu LC20 (Kyoto, Japan). C18 column (2.1 × 50 mm, 1.9 μm) and C18 guard column (inner diameter 2.1 × 5 mm, 1.9 μm) were used to separate ZEN, Z14G, α-ZEL, and β-ZEL on an Agilent Poroshell HPH-C18 column at 40 °C. The elution gradient was applied as follows: 0 min, 80% B; 2 min, 80% B; 10 min, 30% B; 12 min, 30% B; 14 min, 80% B; 16 min, 80% B. The mobile phase consists of (A) acetonitrile containing 0.1% formic acid and (B) double-distilled water containing 0.1% formic acid, with a flow rate of 0.3 mL/min. The injection volume was 5 μL. The mass system was performed using a QTRAP 5500 mass spectrometer (AB SCIEX, Framingham, MA, USA). MS/MS detection was operated in both positive and negative ESI modes. The equipment parameters were as follows: ion spray temperature (TEM): 450 °C; ion spray voltage (IS): −4500 V (in MRM−mode)/5500 V (in MRM+ mode); ion source gas1 (GS1): 55 psi; ion source gas2 (GS2): 55 psi; curtain gas (CUR): 35 psi; collisional activated dissociation (CAD) gas: medium level. The detection parameters of ZEN, Z14G, α-ZEL, and β-ZEL are shown in Appendix A. Analyst software (AB Sciex, Redwood city, CA, USA., version 1.6.0) was used for data analysis.

### 5.6. Statistical Analysis

Statistical analysis was performed with GraphPad Prism 8 statistical software. Biological replicate measurements were taken from distinct biological samples. The data are expressed as mean ± SD (standard deviation). The comparison between groups was based on one-way ANOVA, and the test level α = 0.05. *p* < 0.05 was considered statistically significant.

## Figures and Tables

**Figure 1 toxins-14-00458-f001:**
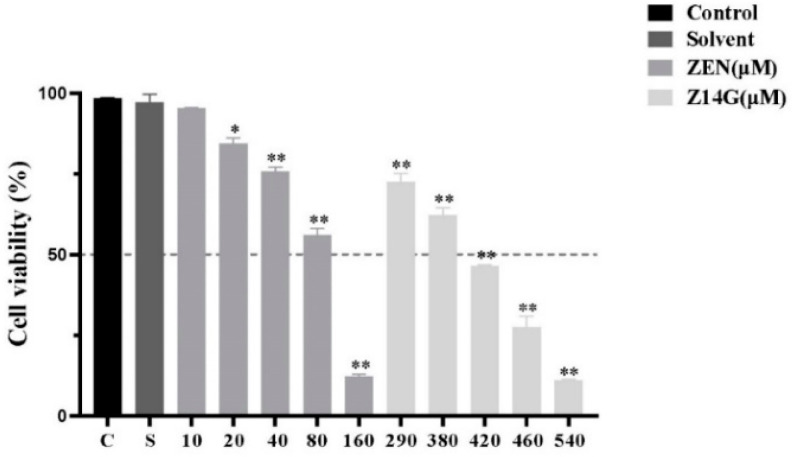
The effect of ZEN (10–160 μM, 24 h) or Z14G (290–540 μM, 24 h) on the cell viability of KGN cells (MTT assay) (*n* = 6, mean ± S.D) * *p* < 0.05, ** *p* < 0.01 vs. Control.

**Figure 2 toxins-14-00458-f002:**
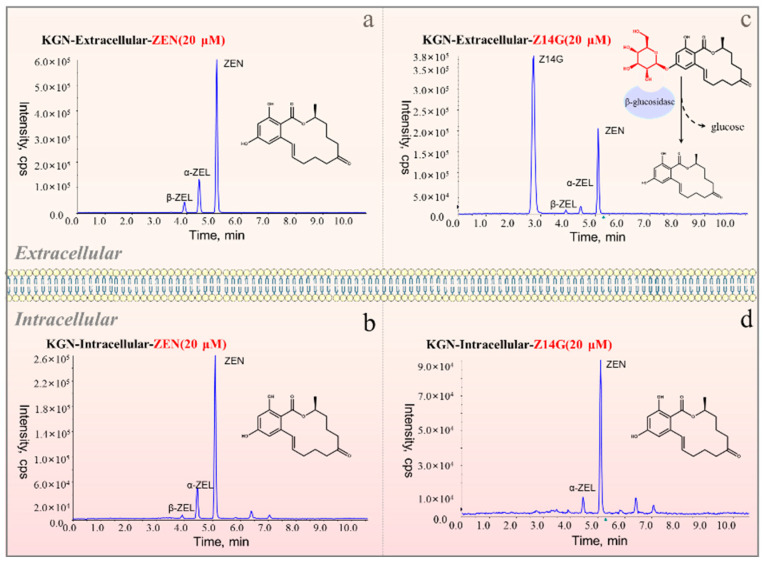
The total ion chromatograms of ZEN (20 μM, 24 h) or Z14G (20 μM, 24 h) in KGN cells. (**a**) ZEN-Extracellular, (**b**) ZEN-Intracellular, (**c**) Z14G-Extracellular, (**d**) Z14G-Intracellular (*n* = 6).

**Figure 3 toxins-14-00458-f003:**
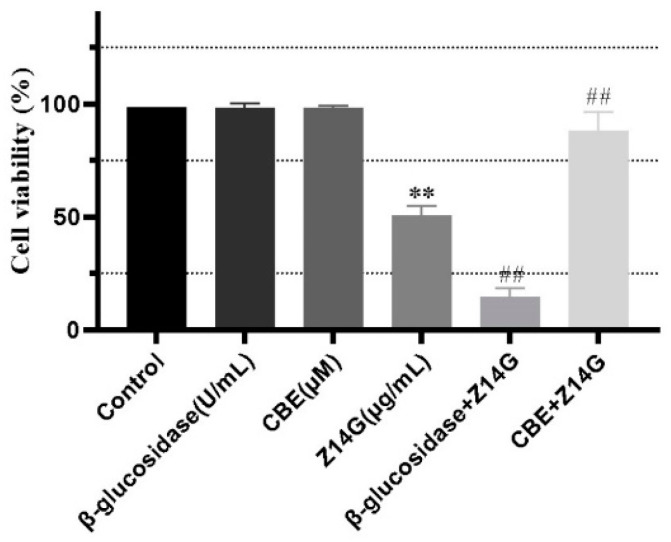
The effect of β-glucosidase (100 U·mL^−1^, 24 h) + Z14G (420 μM, 24 h) co-administration or CBE (100 μM, 48 h) pretreatment + Z14G (420 μM, 24 h) on the cell viability of KGN cells (MTT assay) (*n* = 6, mean ± S.D) ** *p* < 0.01 vs. Control, ## *p* < 0.01 vs. Z14G.

**Figure 4 toxins-14-00458-f004:**
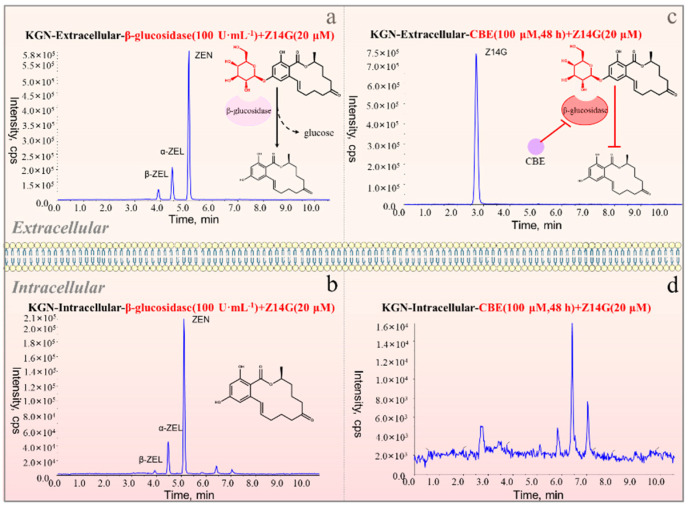
The total ion chromatograms of β-glucosidase (100 U·mL^−1^, 24 h) + Z14G (20 μM, 24 h) co-administration or CBE (100 μM, 48 h) pretreatment + Z14G (20 μM, 24 h) metabolism in KGN cells. (**a**) β-Glucosidase + Z14G-Extracellular, (**b**) β-Glucosidase + Z14G-Intracellular, (**c**) CBE + Z14G-Extracellular, (**d**) CBE + Z14G-Intracellular (*n* = 6).

**Figure 5 toxins-14-00458-f005:**
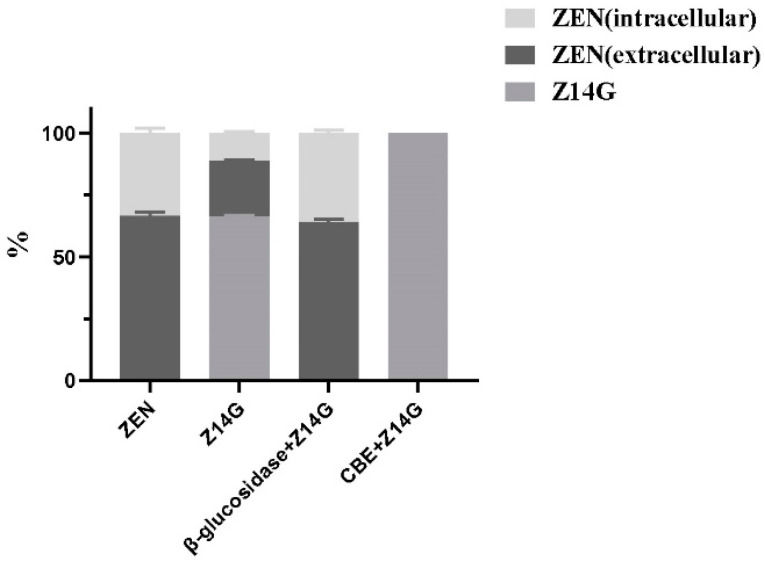
Metabolism of ZEN (20 μM), Z14G (20 μM), β-glucosidase (100 U·mL^−1^, 24 h) + Z14G (20 μM, 24 h) co-administration or CBE (100 μM, 48 h) pretreatment + Z14G (20 μM, 24 h) in KGN cells (*n* = 6, Mean ± S.D).

**Figure 6 toxins-14-00458-f006:**
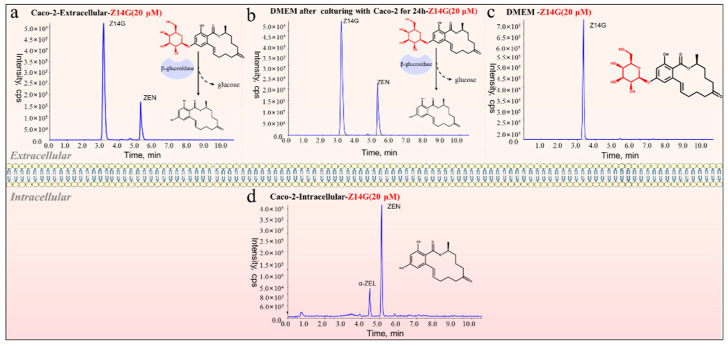
The total ion chromatograms of ZEN (20 μM, 24 h) or Z14G (20 μM, 24 h) in Caco-2 cells. (**a**) Z14G-Extracellular, (**b**) DMEM after Caco-2 cultured for 24 h + Z14G-Extracellular, (**c**) DMEM + Z14G-Extracellular, (**d**) Z14G-Intracellular (*n* = 6).

## Data Availability

Not applicable.

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
