# Peer review of "Zearalenone-14-Glucoside Is Hydrolyzed to Zearalenone by β-Glucosidase in Extracellular Matrix to Exert Intracellular Toxicity in KGN Cells"

_toxins, 2022, doi:10.3390/toxins14070458_

Round 1

Reviewer 1 Report

The present manuscript is dealing with zearelone-14-glucoside, a so called modified or masked mycotoxin of zearalenone. On the one hand the cytotoxicity and metabolization in human ovarian granulosa cells is investigated, on the other hand Caco-2 cells, a human colon adenocarcinoma cell line, were used. Although the data are interested, my biggest concern are the applied concentration ranges which are way too high in my opinion. Furthermore, the manuscript lacks a few important references and needs some general improvement (e.g., mix-up of Figures): Please find below my more detailed comments.

I suggest including the following highly relevant literature at appropriate positions

https://pubmed.ncbi.nlm.nih.gov/30019167/

https://pubmed.ncbi.nlm.nih.gov/27921366/

https://www.ncbi.nlm.nih.gov/pmc/articles/PMC5331435/

https://pubmed.ncbi.nlm.nih.gov/25935671/

https://onlinelibrary.wiley.com/doi/full/10.1002/cche.10360

Please revise the manuscript with regard to the English language:

e.g., line 27: “can combine with” should be replaced e.g., by “that some mycotoxins can be modified with substances”

e.g., line 38; The article “a” should be removed in front of “deglycosylation”.

e.g., line 123: It should be “an Agilent” (instead of “a Agilent”).

e.g., line 1857186: It should be “was not detected intracellular” (twice).

e.g., line 155: it should be “a previous study has shown”.

e.g., line 162: it should be “than that of beta-ZEL” (with “of”)

e.g., line 176: it should be “In the biomedical field” (with the article)

Please comment why you used such high concentrations. 420 µM are 201 mg/L. These concentrations will never be reached in the intestinal system or in ovaries taken into consideration the comparable low concentrations in the middle µg/kg range in food. I also do not understand why you used far higher concentrations for Z14G compared to ZEN which definitely occurs in higher concentrations. Furthermore, please could you comment why you used different concentrations in the cytotoxic and cell metabolism experiments. Thanks!

Specific comments:

Abstract:

Line 5: It should be “zearalenone-14-glucoside” with small starting letters.

Introduction:

Line 6/7: I suggest to rephrase the first two sentences a little bit, since two times “extensive/extensively” are used. Maybe one should be replaced.

Line 37: The dot after “et al.” is missing.

Line 42: I suggest to cite the consolidated version of this regulation or at least mention also the appendments of the 1881/2006 document. The values you report here are actually laid down in the first amendment:

Consolidated version: https://eur-lex.europa.eu/legal-content/EN/TXT/HTML/?uri=CELEX:02006R1881-20220503&from=DE

Line 58: I do not understand? Why is it important to study ZEN, when estrogen is considered carcinogenic? Yes, it is important to get more information about ZEN, but I cannot follow this line of argumentation.

Materials and Methods
Please check, but as far as I know, in case of the MDPI-journal toxins the Materials and Methods are placed at the end of the article after Results and Discussion. Please follow the author guidelines.

Line 69: I do not understand why you added “Isoreg” and “Yuanye” after the compound purchase. The companies are stated anyway.

Line 81: “human” should be written with a small starting letter. Furthermore, please specify which Caco-2 clone you are using since several one exist.

Line 87: Why did you use “methanol” and not ethanol? Ethanol is far less toxic than methanol and a concentration of 1% seems to be quite high.

Line 92: What I do not quite understand why the group of CBE+Z14G does not include beta-glucosidases. What is the purpose of providing an inhibitor, when the enzyme is not added.

Line 93: How often did you repeat this experiment? You only stated it for point 2, but not for point 1.

I miss here information about the cultivation and incubation conditions of Caco-2 cells. How many cells were seeded?

Line 106: First of all, I assume that it was not “just” DMEM you used, but that it contained also FKS and Pen/Strep. Please make it clear to the reader. Secondly, it should be “phenol-red free” and not “red phenol-free”.

Line 116: Please provide the centrifugation speed in g-force instead of “rpm” – otherwise you have to specify the specific centrifuge including rotor you used.

Line 123: The general convention is that in reversed phase chromatography, eluent A is the aqueous and eluent B the organic solvent. Please consider to change it.

Line 127: a flow rate of 0.3 µL/min seems a little bit small for this column. Do you mean 0.3 mL/min? Please check.

Results:

Please elaborate why you used exactly these two cell lines.

Line 142: What do you mean with “unsuitable cell types”. Could you please elaborate this. Thanks.

Line 151: There is a mix up of Figure 1 and Figure 2, the figure captions are correct, but the wrong figure is inserted. Please exchange them.

Figure 1: Is it just an optical illusion or does the column of the control does not reach 100% which would be strange since it was set to 100.

Figure 2: The structure of ZEN is wrong, the hydroxygroup at position C14 is missing in all panels, this is also true for Figure 4.

Figure 2 and 4: Please explain what is displayed here? Do we see the total ion chromatogram or extracted ion chromatrograms? Have you estimated the concentrations?

In my opinion, Figure 3 and Figure 5 could be a little bit smaller – no need to be that big.

Line 209: I am sorry, but I don not understand what you did. You added Z14G to pure DMEM medium and then to DMEM-medium which had been incubated with Caco-2- cells? This is not mentioned at all in the materials and methods section.

References: Currently, the continuous number is displayed twice. Please correct.

Furthermore, please follow the journal guidelines for providing the references.

Ref 21: The world cancer report is from 2020 as far as I know.

Supplementary material:

The formula provided for the recovery lacks the multiplication with 100 to get the result in percent.

Table S1: Please explain the used abbreviations: Q1, Q2, Rt, DP, CE, CYP. I know what they mean since I am familiar with the Sciex nomenclature, but some readers might not know it.

In my opinion it is not necessary to state the Rt twice since it has to be the same for both transitions.

With regard to “DP” it is quite strange that there sometimes is a quite huge difference between the two transitions. The DP is a parameter of  the precursor, so it should be the same for the quantifier and the qualifier.

Reviewer 3 Report

I have enjoyed reading this article, dealing with cytotoxicity of zearalenone-14-glucoside and its toxic release mechanism. The manuscript is well prepared, providing sufficient background clarification. Perhaps few more examples of masked (biologically modified) mycotoxins could be stated, at least deoxynivalenol-3-β-glucoside which is later mentioned in the text. The conducted study experiments are concisely described and results very nicely presented in figures. Determination parameters and validation data of used UHPLC-MS/MS method are also provided (in supplement). English language and style are satisfying and manuscript perspective can be easily understood. Used references are properly cited and mainly current. The manuscript contains publishable results indicating that, different from previous findings, zearalenone-14-glucoside is only extracellularly hydrolysed to its parent form zearalenone by β-glucosidase which then enters the cell to exert its toxic effects.

There are only few minor observations to be addressed before publication.

Line 19 and elsewhere – If I understood right, it is suggested that there should be established maximum permitted level not only for zearalenone, but also for sum of zearalenone and its metabolites? If so, consider replacing limit standard with legislation limit or some other term for better understanding.

Line 116 – Space lacking. Check also through text for similar mistakes.

Line 128 –QqQLIT should rather be replaced with QTRAP 5500 mass spectrometer (AB SCIEX).

Abbreviation explanation missing in some lines, please check: 138 -x ± SD, 223 - MCF-7 cells, 224 - JNK and P38 MAPKs, 233 - GI -tract (maybe add abbreviation in line 31), 260 – TDI.

Round 2

Reviewer 1 Report

Dear authors,

Thank you very much for revising your manuscript according to the reviewer suggestions.

It was very important that you clarified the purpose of the scientific research. However, could you please also add a few sentences to the rationality behind the chosen concentrations. You explain it in the response to Reviewer #1, but not in the manuscript itself. Thank you very much.

Line 11: “human” should be written with a small starting letter.

Line 26: There is a space missing between “[1].” and “Previous”.

Line 30: It should be “so they are named masked mycotoxins” (plural since you refer to “mycotoxins”.

Line 126: There is a dot missing after “vs.” (twice).

Line 150: Please consider revising Figure 6. It is confusing for the reader that in the Figure you write extracellular and intracellular and then Subfigure “d” is also extracellular.

Line 241/242: There should be a space between the number “24” or “48” and “h”. (same in line 252, 253) – Please check the whole manuscript.

Line 242: This sentence is not complete: “On CPE+Z14G administration group”.

Line 244: Are those biological or technical replicates? Hence performed on different days on different passages or on the same day with the same passage? This is also true for the other experiments.

Line 252: “which culture by Caco-2 for 24 h” is not an English expression, please revise (used several times)-

Line 259: The “4” in “5x104” should be superscripted.

Line 260: It is no longer “2.2-(1)”, but “5.2.-(1)”.

Line 263: “Glycine” should be written with a small starting letter.
